# UAV IoT Framework Views and Challenges: Towards Protecting Drones as “Things”

**DOI:** 10.3390/s18114015

**Published:** 2018-11-17

**Authors:** Thomas Lagkas, Vasileios Argyriou, Stamatia Bibi, Panagiotis Sarigiannidis

**Affiliations:** 1Computer Science Department, The University of Sheffield International Faculty, CITY College, 54626 Thessaloniki, Greece; t.lagkas@sheffield.ac.uk; 2Department of Networks and Digital Media, Kingston University, Surrey KT1 2EE, UK; vasileios.argyriou@kingston.ac.uk; 3Department of Informatics and Telecommunication Engineering, University of Western Macedonia, 50100 Kozani, Greece; sbibi@uowm.gr

**Keywords:** security, privacy, drones, IoT, UAV

## Abstract

Unmanned aerial vehicles (UAVs) have enormous potential in enabling new applications in various areas, ranging from military, security, medicine, and surveillance to traffic-monitoring applications. Lately, there has been heavy investment in the development of UAVs and multi-UAVs systems that can collaborate and complete missions more efficiently and economically. Emerging technologies such as 4G/5G networks have significant potential on UAVs equipped with cameras, sensors, and GPS receivers in delivering Internet of Things (IoT) services from great heights, creating an airborne domain of the IoT. However, there are many issues to be resolved before the effective use of UAVs can be made, including security, privacy, and management. As such, in this paper we review new UAV application areas enabled by the IoT and 5G technologies, analyze the sensor requirements, and overview solutions for fleet management over aerial-networking, privacy, and security challenges. Finally, we propose a framework that supports and enables these technologies on UAVs. The introduced framework provisions a holistic IoT architecture that enables the protection of UAVs as “flying” things in a collaborative networked environment.

## 1. Introduction

The applications of unmanned aerial vehicles (UAVs) are diverse, including areas related to civilian, military, commercial, and governmental sectors [1,2,3,4,5]. Examples include environmental monitoring (e.g., pollution, health of plants, and industrial accidents) in the civilian sector. In military and governmental areas, we mainly have surveillance and delivery applications aiming to acquire or provide information at locations after a disaster or attack, and to distribute medicine or other essential items. Commercial applications are focused on delivering products and goods both in urban and rural areas. UAVs, since they are dependent on sensors, antennas, and embedded software, are considered as part of the Internet of Things, providing a two-way communication for applications related to remote control and monitoring [6].

The Internet of Things (IoT) constitutes a rapidly emerging cutting-edge environment in which the focal concept lies in the orchestration of a large variety of smart objects in such a way that they can be utilized and operated globally, either directly by users or by special software that captures their behavior and objectives. IoT enables objects to become active participants of everyday activities, with numerous promising applications through various communication technologies in the context of the “smart-city” vision [7]. It is estimated that around 25 billion uniquely identifiable objects are expected to be part of this global community by 2020. These projections are expected to substantially increase with the introduction of 5G technologies and networks.

IoT objects are becoming more complex, heterogeneous, and highly distributed [8]. This transformation comes with a cost: the IoT, as a fusion of heterogeneous networks, not only involves the same security problems with sensor networks, mobile communication networks, and the Internet, but also brings along specific privacy-protection challenges. As part of heterogeneous networks, things have to support advanced security concepts, such as authentication, access control, data protection, confidentiality, cyber-attack prevention, and a high level of authorization [9]. These security and privacy challenges are different from traditional Internet security issues, since the IoT presents unique features in handling and dealing with external and internal threats. In this context, a regulatory framework is needed for setting and applying rules and policies in commercial objects. This framework should provide regulation rules and procedures that all commercial things should pass for receiving a security and privacy license in terms of connectivity and intelligence, actuation, and control features.

In light of the aforementioned remarks, this paper:Overviews new UAV application domains enabled by IoT and 5G technologies.Analyzes the IoT sensor requirements for drones.Summarizes the privacy and security challenges of UAV applications.Overviews solutions for fleet management over aerial networking.

Furthermore, to address IoT security and privacy challenges for drones, an advanced framework for end-to-end security and privacy prevention in real, market-based dynamic IoT environments is introduced. The proposed framework includes cutting-edge holistic approaches for advancing the current security and privacy level into a robust, resilient, and high-protected trusted environment. The framework supports multilevel and multidomain defence mechanisms in protecting IoT objects (i.e., UAVs) from spoofing, signal-jamming, and physical attacks, RF and mobile-application hacking, protocol abusing, and firmware hacks/sabotage. Privacy preservation is accomplished by effective ‘crowd of things’ strategies, where the anonymity of the users and the information that the UAV carries are ensured. Vision techniques are considered as aiming to enhance the security of IoT by supporting computer-vision and machine-learning solutions.

In summary, the main contribution of this work is, on the one hand, conducting a targeted review that focuses on security issues and promising solutions associated with the inclusion of UAVs in the IoT ecosystem, considering the special characteristics of such devices and the related cutting-edge technologies. On the other hand, a new framework that involves UAV-specific security extensions is presented for addressing the identified issues, along with ambitious real-world use cases.

The rest of this paper is organized as follows. In Section 2, we discuss aspects of using UAVs for wireless networks, review prior UAV–IoT frameworks, overview IoT Sensors for UAVs over 5G networks, discuss security and privacy issues for drones, and analyze protection mechanisms focusing on aerial networks and fleet-management systems. In Section 3, we describe the proposed framework focusing on protecting drones. Suggestions and evaluation requirements are presented in Section 4, and we finally conclude the paper in Section 5.

## 2. Overview on UAVs as Members of IoT

### 2.1. UAVs for Wireless Networks

#### 2.1.1. Use Cases for Wireless Networking with UAVs

The use of UAVs as key entities of next-generation wireless networks constitutes one of the most promising applications of the corresponding technologies. A number of promising use cases are thoroughly detailed in Reference [10] and presented below.

UAV-carried flying base stations that complete heterogeneous 5G systems to enhance the coverage and capacity of existing wireless access technologies.UAV-based aerial networks that allow reliable, flexible, and fast wireless connections in public-safety scenarios.UAVs that support terrestrial networks for disseminating information and enhancing connectivity.UAVs as flying antennas that can be deployed on demand to enable mmWave communications, massive MIMO, and 3D network MIMO.UAVs that are used to provide energy-efficient and reliable IoT uplink connections.UAVs that form the backhaul of terrestrial networks to allow agile, reliable, cost-effective, and high-speed connectivity.UAVs able to cache popular content and efficiently serve mobile users by following their mobility patterns.UAVs that act as users of the wireless infrastructure for surveillance, remote-sensing, and virtual-reality cases, and package-delivery applications.UAVs that collect vast amounts of city data and/or enhance cellular network coverage in a smart-city scenario.

#### 2.1.2. UAV Types and Classifications

Different types of UAVs with distinctive characteristics, such as supported altitude, speed, and energy autonomy, are suitable for different applications. Generally, UAVs are classified according to their supported altitudes into Low-Altitude Platforms (LAP) and High-Altitude Platforms [11]. Furthermore, UAVs can be classified into rotary-wing and fixed-wing. The former are appropriate for cases that require UAVs that can remain at steady positions, whereas the latter are suitable for applications that demand UAVs travelling at high speeds and covering large distances [12]. In an IoT environment, due to the limited energy capacity of the participating devices, suitable LAP UAVs of the rotary-wing type can be efficiently and dynamically positioned to allow IoT devices to transmit with minimum power. A related framework towards this direction is introduced in Reference [13], while authors in Reference [14] introduce a resource-allocation scheme for improving energy consumption at cluster heads that use aerial base stations.

#### 2.1.3. Interference Management, Deployment, Path Planning, and Energy Consumption of UAVs in IoT Networks

The use of UAVs as flying relays for IoT networks has numerous advantages, such as energy conservation and reliability; however, there are also some significant challenges that need to be addressed. Among those challenges, interference management, UAV deployment, and path planning are considered of major importance. The authors in Reference [15] propose and analyze an efficient deployment scheme for multiple UAVs using circle parking theory.

Regarding interference management, the findings revealed that UAVs’ altitude needs to be adjusted according to the coverage requirements and the beamwidth of their directional antennas. A related work presented in Reference [16] concluded on the optimum placement of UAVs as relay nodes that the decode-and-forward approach outperforms the amplify-and-forward one. A new heuristic algorithm for 3D UAV deployment was introduced in Reference [17], which minimizes the number of required UAVs to keep a specific level of service quality. To mitigate interference, the authors suggest lowering the altitude, but there is an obvious tradeoff between this and coverage. Similarly, the authors in Reference [18] analyze the tradeoff between delay and coverage, as far as the number of UAV stop points is concerned.

As far as path planning is concerned, it is directly related with trajectory optimization. In general, finding the optimal flight path for a UAV is considered a challenging goal, since it is affected by multiple factors, such as energy limitations, flight time, and obstacle avoidance. Hence, as explained in Reference [10], path planning is usually approached as an optimization problem with various objectives depending on the criterion of interest.

Energy consumption, in particular, constitutes a critical issue for the deployment and mobility of UAVs. Because of their limited battery capacity, UAVs are not typically able of providing for long continuous wireless coverage in scenarios such as IoT networking. Their energy autonomy is highly affected by the UAV role and flight path, weather conditions, etc., and actually constitutes the main constraint for UAV adoption in many cases. There are several recent research endeavors toward improving UAV energy efficiency, focusing on various aspects, such as trajectory optimization [19], co-operative communications [20], energy harvesting [21], and resource allocation [22], et al.

### 2.2. UAV–IoT Frameworks

Due to UAVs’ high agility, they are now widely accepted as promising members of the IoT vision or even enablers of such a vision. They are capable of offering new value-added IoT services, while they can carry a variety of MTMC devices [23]. In more detail, according to the definition of IoT, “things” are expected to be able to be connected anywhere at any time providing any service. UAVs can fulfil this requirement, thanks to their autonomy, flexibility, and programmability. In this context, a number of UAV-enabled IoT frameworks supporting a variety of practical use cases have been proposed.

Authors in Reference [1] introduced and demonstrated a UAV-based IoT platform for crowd surveillance. The respective platform adopts and applies face recognition techniques and performs efficient offload of video processing to a Mobile Edge Computing (MEC) node, considering the limited processing power and energy capacity of a UAV. The developed testbed collects video-surveillance data and performs face recognition to identify suspicious individuals by utilizing the Local Binary Pattern Histogram (LBPH) algorithm of the Open Source Computer Vision (OpenCV) library. The proposed platform considers central management of a fleet of UAVs through a system orchestrator.

A communication framework for UAVs in urban IoT environments was proposed and evaluated in Reference [24]. It forms a multipath multihop infrastructure that is used to connect the UAVs to the ground control station. The conducted real-world experiments have shown that the introduced framework significantly enhances the control effectiveness and reliability against local congestion. It is noted that the specific work was inspired by the DARPA Hackfest on Software Defined Radios.

In Reference [25], a game-theory-based framework was introduced for allocating resources to UAVs, which enter the IoT ecosystems as platforms that assist terrestrial base stations. The access competition among the UAVs for bandwidth is modelled as a noncooperative evolutionary game. The evaluation of the two designed algorithms showed that Nash equilibrium can be quickly reached.

An optimization framework for aerial sensing in the context of an IoT infrastructure was designed and presented in Reference [26]. The goal is to allow remote users to navigate in specific scenes of interest by using augmented-reality (AR)/virtual-reality (VR) devices over the captured data. The corresponding scenario is likened to virtual human teleportation. The conducted experiments effectively demonstrated the advantages of the proposed methods on visual sensing.

Authors in Reference [27] conceived and presented a new MEC framework for IoT through an air–ground integration approach. Four use cases are presented to show how the proposed air–ground-integrated MEC framework supports high mobility, low latency, and high throughput for 5G applications. Through simulation-based and case-based evaluation, it was shown that the respective framework can support multiple IoT scenarios.

A novel framework for deploying and efficiently moving UAVs to gather information from ground IoT devices is proposed in Reference [28]. This work focuses on the optimal deployment and mobility of UAVs, as well as the optimal clustering of IoT devices, toward minimizing transmission power while retaining reliability. In this manner, it was shown that IoT devices’ energy consumption can be significantly reduced, whereas UAVs can serve as ground devices for longer.

### 2.3. 5G and IoT Sensor Technologies for UAVs

The 5G technology is expected to enhance mobile broadband, enable applications that require ultrareliable very low latency and very high availability networks, improve traffic safety and control, support industrial applications, remote manufacturing, training, surgery, logistics, tracking, and fleet management. It will be utilized for smart agriculture, precision farming, smart buildings, smart meters, support of 4K/8K UHD broadcasting, virtual and augmented reality without range limitations, including homes, enterprises, and large venues offering massive and critical Machine-To-Machine-Type Communications (MTMC) [29]. This type of device communications can be integrated with typical Human-Type Communications (HTC) through suitable gateways in the context of a 5G architecture, as presented in Reference [30] and illustrated in Figure 1.

Other 5G use cases that are linked to drones involve automation and robotics. Drones with the support of 5G networks would be able to offer a wide variety of tasks and applications providing new benefits to a wide range of industries. Among the top use cases of 5G-enabled drones, we consider applications related to construction, agriculture, insurance claims, police, fire, coast guard, border control, journalism, news, utilities, filmography, and logistics. All these applications will be feasible since autonomous and beyond line-of-sight control will be supported. In order to enable these use cases, specific minimum requirements for aerial vehicles are essential in terms of equipment and sensors.

The main types of sensor technologies that are supported and are a part of drones today can be separated into three main categories: (a) flight control, (b) data acquisition, and (c) communication sensors.

#### 2.3.1. Flight Control Sensor for Internal State Evaluation

Accelerometers are used to determine position and orientation of the drone in flight. One type of technology senses the micromovement of embedded structures in an integrated circuit. Thermal sensing is another technology used in accelerometers, which does not include any moving parts but instead senses changes in the movement of gas molecules passing over a small integrated circuit [2]. Drones and UAVs manage to maintain flight paths and directions using inertial measurement units combined with GPS. The Inertial Measurement Units utilize multiaxis magnetometers (available in one to three axes). A magnetometer is basically a magnetic compass that can measure the magnetic field of Earth. This mechanism helps in determining the direction of a compass and, consequently, of the drone, which is estimated with respect to the magnetic North. The flight-control system, in order to maintain level flight, obtains input from tilt sensors combined with accelerometers and gyroscopes. This is an essential element for UAVs, especially when the applications require high level of stability (e.g., surveillance, delivery, etc.). In certain drones we have Engine Intake Flow and current sensors [31]. These UAVs are powered with gas engines to effectively monitor the air flow and sensors to estimate the proper fuel-to-air ratio at a specified engine speed aiming to reduce emissions and the overall consumption. Current sensors are available in drones to monitor and optimize power consumption and detect faults with motors or other areas of the system.

#### 2.3.2. Data-Acquisition Sensors

Drones are equipped with several sensors to capture information and data that are required to perform certain tasks. Depending on the application, the payload sensor suite can be arranged during the development of the drones.

For military use cases, UAVs may be equipped with high-end electro-optical sensors, and radars for airborne systems providing resolutions from submillimeters to a few centimetres.In surveillance and monitoring applications, we can have sensors at the lower end of the spectrum, such as low- or high- (e.g., 4K) resolution RGB (Red Green Blue) cameras, NDVI (Normalized Difference Vegetation Index) cameras for precision farming, LIDAR (Light Imaging, Detection, And Ranging) for simultaneous localization and mapping, and ultrasonic sensors for sense and obstacle-avoidance methods.We can also have hyperspectral depth and thermal sensors [32]. Applications that monitor environmental and weather conditions and are deployed in disaster relief and management require sensors to measure or detect liquefied petroleum gas (LPG), butane, methane (CH4), hydrogen, smoke, oxygen, temperature, and humidity.

In all these applications, the IoT sensors collect data in real time and are either processed on board if enough power is available or transmitted to a base station.

#### 2.3.3. Communication Systems

Managing and controlling tasks for UAVs are performed through communication systems and networks [33]. In the case of multiple drones, technologies are required to allow them to communicate with each other for safety reasons. There are different types of communications, and some of the main types used in UAVs are listed in Table 1. An extensive list of network protocols and communication techniques for generic IoT devices can be found in Reference [34]. Based on the coverage range, available data rates, and latency specifications, it is evident that 5G technology would impact drones’ communications, enabling several worldwide applications. In such a conceptual model, UAVs can form infrastructureless dynamic network segments of the IoT architecture, which are interconnected to the core network for the provision of demanding services, such as surveillance multimedia streaming [35].

### 2.4. Security for UAVs over IoT

Security provision in UAVs as part of the IoT environment is a complex task that requires the efficient integration of various techniques that are associated with different aspects of IoT networking and UAV operation. In the following two subsections, the security and privacy components for such an endeavor are discussed in detail. The main concept of the followed approach is the application of UAV-specific security extensions to various IoT technologies and security techniques, which can enable the required integration.

#### 2.4.1. Security Component

IoT-based UAVs is a complex paradigm in which people and machines interact with the technological ecosystem based on smart objects through complex processes [36]. When studying security aspects in the IoT technological ecosystem, the study focused on four main layers that are, from top to bottom: the application, network, and link/adoption layers, and the perception. Figure 2 presents the IoT security techniques of each layer, along with UAV-oriented extensions (at the right side).

In the application layer, security mechanisms such as the Constrained Application Protocol (CoAP) [37], access control and user application security are included to enable secure messages and minimal configurations using the RESTful library, filtering and perimeter security with the Simple Object Access Protocol (SOAP) and data filtering and cloud support using application firewalls and Intrusion Detection Systems (IDS) [38]. The main UAV extensions in this layer are related with vision-based enhancements that support self-protection and path/destination identification by using video-processing and computer-vision techniques [39,40]. Novel solutions using deep learning and a combination of deep and shallow networks can be applied to enhance the security of mobile things.

In the network layer, mechanisms like IP Security (IPSec), secure routing, and transport security were suggested in Reference [36]. These mechanisms offer IP payload confidentiality and integrity using IPSec and the Encapsulated Security Payload (ESP) protocol. Trusted IoT drones and low-energy protocols can also be available with the IPv6 Routing Protocol for Low-Power and Lossy Networks (RPL) according to the work in Reference [41]. The Datagram Transport Layer Security (DTLS) protocol, which is, in practice, TLS with added features to deal with the unreliable nature of User Datagram Protocol (UDP) communications, and Rivest Shamir Adelman (RSA) algorithms provide enhanced confidentiality, integrity, and authentication. These mechanisms, according to the work in Reference [42] can timely inform about potential threats (e.g., blackhole, wormhole, Sybil attack, and hello attack) in “flying” things while operating. Moreover, as UAV security extensions, novel routing and name service mechanisms can be developed based on the Named Data Networking (NDN) infrastructure, which is an evolution of the IP architecture that generalizes the role of this thin waist, such that packets can name objects other than communication endpoints [43]. NDN extensions can provide robust object identification and ensure the integrity of the things’ records used in the naming architecture, in order to defend against DNS, Man-in-the-Middle (MiM), and identification attacks using DNS Security Extensions (DNSSEC). Additionally, power-aware routing extensions can enhance drones’ energy autonomy and increase flying time.

In the link/adoption layer, important security services have to be applied in the 6LoWPAN adoption layer, including end-to-end security, continuous authentication schemes [44,45], trust verification/validation, and key management mechanisms [37]. To provide robust communication and endpoint security, secure and trusted channel protocols were proposed in Reference [46]. Furthermore, group key management techniques were considered in Reference [38], aiming to enable multicast communication using CoAP. To further strengthen the IoT drones, a trusted computing framework and execution architecture, were proposed in Reference [47]. Secure connections are enabled with connection firewalls and Intrusion Protection Systems (IPS) that can deeply inspect network packets. In addition, firewalls, intrusion detection systems, and colocation proofs were suggested in Reference [48] that are necessary against routing attacks, such as selective forwarding, sinkhole, and wormhole attacks [44].

The primary security threats in the perception layer are physical damages, such as vandalism and weather challenges. Therefore, hardware mechanisms such as tamper-protection and -detection systems have to be employed, as was suggested in References [36,49]. In the perception layer, security is to ensure that only authorized users can have access to sensitive data that are produced by physical objects, and that’s why authentication and authorization policies need to be defined. Authentication issues are addressed by combining light public and symmetric key mechanisms. According to Reference [50], RFC 6698 TLSA and Numberg’s scheme are used to ensure data authenticity. The problem of authorization is addressed by combining open protocols such as Oath, HTTPS, U2IoT, and OpenID Connect [51]. In addition, lightweight cryptography and digital signature schemes are needed to ensure secure end-to-end communications. Light versions of existing algorithms such as the Advanced Encryption Standard (AES), the RSA, and Elliptic Curve Cryptography (ECC) can be customized to be applied on mobile things.

#### 2.4.2. Privacy Component

Privacy protection in the IoT domain is divided into two main directions: (a) protection of collecting, acquiring, and distributing sensitive information such as faces, body silhouettes, objects, and license plates, by “flying” things such as drones; and (b) protection of observing and eliciting patterns, such as the number, duration, and diversity of connections, all of which can be used as signatures of IoT devices. The former direction is also referred as “user privacy”, while the latter one is known as “thing privacy”.

Privacy protection is a challenging goal in many research fields where measuring devices take place. For example, in Reference [52] a wide range of sensors measure various types of information; however, the concept fully protects the privacy of all registered users. To ensure anonymity, the group signature and data of all devices are communicated via TLS protocol with an authentication mechanism. The integrated group signature method appears to be efficient to protect the identity of the measuring devices. A group signature is used to encrypt all data in order to avoid the reception of dummy data by the reporting service. Similarly, in Reference [53], the proposed work provides identity confidentiality of both parties (the end user who requests data and the owner of the mobile device). In that manner, the end user cannot associate the mobile device and measured data.

Current solutions for “user privacy” in “flying” things are focused on filters aiming to remove sensitive information. Most of these systems are based on computer vision analyzing video content [54]. In the case of mobile things, such as drones, since they can get close to targets and capture the same scene from different points of view, they are able to gather sensitive personal data, adding a new dimension to issues related with privacy and protection solutions. Several state-of-the-art privacy filters were introduced, aiming also to keep a balance between privacy issues and surveillance effectiveness. Privacy filters include simple filters such as pixelization, masking blurring, as well as more advanced morphing [55] and reversible warping [56] filters. Recently, several researchers [57,58,59,60] proposed new features and approaches for drone-based surveillance that affect visual privacy. Regarding the acquired audiovisual data, a number of approaches have been proposed that employ encryption techniques [61,62,63]. However, such techniques cause the corruption of large parts of the original image, making the intelligibility task practically impossible. Other solutions use reversible scrambling applied in the compression-specific domain of a particular video format [64,65,66], improving the visual result, but heavily depending on the employed compression algorithm.

“Thing-privacy” breaches arise because it is still possible to observe patterns such as the number, duration, and diversity of connections, all of which can be used as the signatures of IoT devices. Existing schemes that preserve privacy in identity management are mostly centralized solutions [67]. Data provenance can provide data trackability, not just for data privacy but also for data-quality assurance and data-management transparency [68,69].

### 2.5. Protection for UAVs

One of the main challenges in the design of co-operative applications involving Multiple UAVs (Multi-UAVs) is the formulation of a network that can provide connectivity among the different types of employed vehicles, protecting at the same time the vehicles and the fleet mission from failures [70]. Therefore, it is very important to protect the life cycle of a fleet by (a) establishing and maintaining flexible aerial networks, and (b) by applying effective fleet-management techniques.

#### 2.5.1. Aerial Networking

Multi-UAV networks should demonstrate highly dynamic behavior on complex operating scenarios baring all the constraints related to energy consumption and connectivity that may jeopardize the fleet mission. Research on aerial networks focuses on the definition of routing protocols that ensure quick communication recovery by employing flexible aerial nodes and energy-consumption management techniques that enable increased flight lifetime.

At the moment, configurations like Mobile Ad-hoc NETworks (MANETs) that are destined for mobile-device communications, Vehicle Ad-hoc NETworks (VANETs) that mainly consider land vehicles, and Flying Ad-hoc NETworks (FANETs) are adopted, each of them presenting certain advantages and disadvantages in context-specific deployments [71]. All of the above adopt ad hoc routing protocols that allow the exchange of data from one node to another without any direct links. There are three main types of routing protocols, namely, proactive, reactive, and geographic [70].

Proactive protocols [72] incorporate tables for each node that are periodically updated to store routing information for all other nodes of the topology. The main advantage of proactive protocols is that the tables of each node contain up-to-date information on routes due to continuous message exchanges; a fact, though, that causes bandwidth constraints.Reactive protocols [73] search and store routing paths between two nodes only when the need for communication between them arises. These types of protocols present the advantage of consuming low bandwidth, but there are many cases, specifically in large topologies, where route-path calculation is very slow, causing high latency.Geographic protocols [74] assume that the source node is aware of the geographic position of the receiving node and therefore sends the message directly without the need for searching for a route path. This protocol is very effective in terms of latency, bandwidth, and throughput, though localization information should be available. Such information can be very challenging to obtain in GPS-denied environments; however, this is quite unlikely to occur in the case of FANETs, while signal-based tracking methods (such as the one introduced in Reference [75]) can be additionally adopted.

Energy conservation in UAV networks is also very important in protecting network continuity and increasing the carried payload. In terms of networking, saving energy in UAVs can be achieved [76] by (a) data reduction, (b) network coding, and (c) energy-efficient routing. Data reduction is possible by adopting aggregation schemes that perform data fusion, combining data derived from nodes on the same path. In this direction, adaptive data sampling is also an option for data-gathering tasks performed by UAVs by adjusting the sampling rate (images, videos, etc.) without compromising the required information precision. Network Coding (NC) is also used to reduce data traffic in broadcast scenarios by sending a linear combination of several packets instead of a copy of each packet.

On the other hand, energy conservation can also be enabled through the appropriate energy-routing mechanisms that take into consideration metrics relevant to power utilization and load distribution. Power utilization is related to the remaining battery of a node, while load distribution refers to the queue status of a node measured as the number of packets received and waiting to be transmitted. To this end, there are three main energy-aware routing protocols: multipath-based protocols, node-based protocols, and cluster-based protocols [70]. Multipath-based protocols balance energy consumption among nodes by alternating forwarding nodes. These protocols discover multiple node-disjoint routes utilizing a cost function based on the hop distances and the energy levels of the nodes, and allocate the traffic rate to each selected route. Node-based protocols do not only consider the shortest paths, but select the next candidate hops based on their residual energy. Cluster-based protocols organize the network into clusters, where each cluster is managed by the cluster head (CH), which is responsible for co-ordination and aggregation operations. Sleep/wake-up protocols that save power by setting as many nodes in idle mode for as long as possible.

#### 2.5.2. Fleet Management

Fleet management in autonomous moving objects is a challenging research field that focuses on defining the optimal formation configuration (positioning, speed, height) of a fleet of heterogeneous aerial objects, including decision making in the case of collisions or accretions [77]. Fleet-management techniques may offer various Levels of Automation (LOA) that can range from fully automated flights (no human involvement) to fully human-operated flights. Many works proposed automation architectures [78,79], but human operations are still usually required [80,81,82].

Fleet-management techniques can be classified into centralized and decentralized schemas [83].

In the centralized schema, a formation manager that can be one of the aerial vehicles of the fleet or a ground-based station [82], acts as a supervisor for all aerial vehicles and manages their topology. Centralized schemas present the advantage that important decisions are performed at a higher level, by centralized high-power computer systems, where humans can also interfere. On the other hand, the major disadvantage of this schema is that it requires frequent ground communications, which can be energy consuming and, in case of disruptions/failures, ground management can cause delays [83].In the decentralized schema [4], each aerial vehicle has a certain freedom in decision making, while the whole formation must be capable of reconfiguring, making decisions, and achieving mission goals. This schema is energy-efficient and presents reduced reaction times, though it may produce conflicting decisions, jeopardizing the fleet formation and, in cases of critical formation updates, it may require ground-control assistance [83].

## 3. Proposed Framework and Methodology

### 3.1. Overall Concept and Methodology

The fundamental concept of the proposed framework to protect drones lies in the fact that mobile semiautonomous devices are expected to enter the IoT architecture as another type of smart devices and, due to their significant impact on several everyday activities (sensitive/critical or not), they require well-established and high-quality security support. On this ground, the proposed framework envisions to provide the necessary security components that would facilitate the process of interconnecting drones and UAVs under the umbrella of the IoT, while exhibiting advanced intelligence and self-management characteristics.

In more detail, considering the use of drones for different types of scenarios as representative application, the current state is actually characterized by separated, isolated, and custom approaches. As illustrated at the top of Figure 3, each operator currently uses their control center to communicate and control drone flight based on their own perceived processes. The adoption of security tools is, in fact, optional, their quality is questionable and not certified, as well as not compatible (in the general case) with security measures applied by other operators. The flight license (wherever required, based on local regulations) is also separate and difficult to verify compliance. Moreover, there are no fleet-management processes and no interdrone communications.

The introduced framework actively inserts drones in the IoT architecture by properly securing them both at device and network level.

At device level, the proposed framework enhances drones with embedded lightweight security, privacy, and safety based on cutting-edge vision-based techniques, which also enable advanced scene/path identification.At network level, drones become part of the IoT architecture and they are accessed/controlled through it. Furthermore, agile communications among drones are enabled, providing self-organizing capabilities that set the basis for innovative features, namely, device registration, flight dynamic monitoring, trust establishment through a distributive reputation point system, enforcement and verification of flight-plan regulations, and extensive fleet management via advanced interoperability.

For achieving the aforementioned security and privacy goals, the proposed framework introduces: (a) a lightweight security toolbox, (b) vision-based solutions, and (c) privacy prevention and anonymity techniques for mobile things.

#### 3.1.1. Lightweight Security Toolbox in “Flying” Things

In the context of the introduced framework, a novel lightweight security toolbox is proposed designed for “flying” things. The security toolbox supports open-source end-to-end security, authentication, and key management mechanisms in the adoption/network layer, firewalls and intrusion detection systems in both the adoption/network and application layers, and access control and selective disclosure in the application layer. The proposed security toolbox is lightweight and flexible since it is embedded in the drone’s firmware as a part of the core software. Furthermore, the novel security framework incorporates physics and deep-learning mechanisms to allow the estimation of anomalies and security threats (e.g., hijacking).

#### 3.1.2. Vision-Based Novel Solutions as Security Enhancement

Another issue addressed in the proposed framework is related to human’s or drones’ safety and self-protection. There are many scenarios (e.g., hackers take control of a drone, damaged and attached drones, etc.) in which drones may be a potential threat for humans or extra safety is required for the sensitive items they may carry. In order to overcome these issues, scene analysis and understanding using computer vision have been considered. Therefore, semantic segmentation is supported in this framework, aiming to offer complete forms of visual scene understanding, mainly for outdoor scenarios.

#### 3.1.3. Privacy Prevention and Anonymity in “Flying” Things

In the introduced framework, we addressed both “user privacy” and “thing privacy” challenges in the “flying” things domain. To this end, user-privacy prevention is obtained by allowing mobile things to collect information that describe a user in detail, but preserving the privacy of the collected data. The “UAV privacy” concept is applied in mobile things supporting further anonymity techniques, such as k-anonymity, group signature, and crowd of things, customized for drone IoTs. A community of mobile things, configured as IP-based drones, are formed to apply anonymous authentication using Anonymous Access Credentials (AACs).

### 3.2. Proposed UAV IoT Architecture

As shown in Figure 4, the introduced architecture supports multiple solutions for security, safety, and privacy for mobile IoT toward the provision of services to different users through a set of distributed systems. In this way, end users can deploy and query their “flying” things, such as drones, in a secure and safe ecosystem respecting existing privacy regulations. In the proposed architecture, we have three main entities: the control center, the distributed systems, and the embedded solutions on the UAVs. The system is based on a drone-to-drone communication utilizing rooting algorithms, while communication with the control center takes place through the Internet. The Control Center (CC) is responsible for the management and orchestration of the proposed ecosystem. For that purpose, the CC shown in Figure 4 is responsible for end-to-end communication with the “things”. It integrates all traditional IoT management elements and novel functional blocks to realize searching, information retrieval, and group instruction administration. Distributed Systems (DSs) are focused mainly on three areas: monitoring, management, and reputation. In more detail, the monitoring system is responsible for the registration processes and information authentication. In the case of the reputation system, we consider ranking functionalities for all drones, whereas the last component provides fleet-management solutions. The third entity represents all components integrated in the “flying” things.

In the proposed architecture, we focused mainly on the security and privacy components supported by vision-based systems. One component targets on security, protecting the drone from attackers aiming to hijack it and take control using network and wireless channel exploitation techniques. A second component is related to the privacy of the payload that incorporates advanced encryption and anonymity solutions. A third computer-vision component is included that provides support and solutions both for security and privacy. This component introduces mechanisms for destination and path verification for security, as well as behavior analysis and scene understanding for privacy and safety, respectively. Finally, components for monitoring and drone-to-drone communication supporting registration and routing algorithms are part of the embedded system on the mobile IoT devices.

### 3.3. Potential Security-Sensitive UAV IoT Applications

The deployment of the proposed UAV IoT framework/architecture for the protection of drones enables the realization of innovative security-sensitive applications. In this subsection, two such applications are presented and discussed.

#### 3.3.1. Power-Line Monitoring

In this case, we exploit topics related to anonymity of IoT mobile devices, vision-based security using scene analysis, secure communications, and safe landing locations. The proposed scenario involves power-line monitoring and inspection, mainly over rural areas, as illustrated in Figure 5. Power-line monitoring is essential for all high- and medium-power operators/distributors. Today, most inspections are carried out with aerial methods with the use of helicopters, while sometimes terrestrial methods are used as well (e.g., ground patrols). Both methods are expensive and time consuming without guaranteeing successful results. Operators spend a lot of time and money to repair damages that were not detected during inspections. According to the proposed application, unmanned multicopters would be equipped with a set of visual, infrared, and localization sensors. All data would be acquired simultaneously, and a flight path would be scheduled based on power-pylon positions. This includes X, Y, and Z co-ordinates that would be used as a reference for the calculation of mission waypoints. A flight route would be designed in a way that multicopter flies several meters above the power pylons, several meters beside them in one direction, and then back to the other side of the power line. This would allow for capturing oblique inspection images of both sides of the power infrastructure, as well as point clouds and nadir images with a large side overlap, resulting in doubled density of point clouds and images suitable for accurate orthophoto building and stereoscopic analyses. After landing, all acquired data and flight logs would be downloaded. They would be analyzed for quality and completeness. The following flights would be planned retaining a buffer in order to allow overlapping with the previous mission to assure continuity of data along the power lines.

#### 3.3.2. Human Blood Delivery

In this case, we exploit topics related to payload privacy and monitoring, advanced security topics, thing anonymity, and embedded security tools. The proposed scenario involves hospitals and medical vans delivering human blood between them, which is an extremely vulnerable and high-demand task, as depicted in Figure 6. Delivering human blood is an assignment of high demand and risk. There are also certain requirements and restrictions imposed regarding temperature, time, and exposure. The payload in this case contains further sensitive and private information that may be vulnerable to external threats and attacks. This delivery scenario would provide solutions for many security and privacy issues, mainly related to the payload, considering approaches to provide a secure and robust routing protocol. It involves precise flights over urban areas following preselected and precalculated paths, payload protocols for security and privacy, and safe landing and dropping of the blood in a specified location (from medical vans to hospitals or vice versa). The multicopters, which would be equipped with a high-resolution camera, localization sensors, and cargo container, would fly to the predefined destination using a direct root, minimizing the required time. Regarding the container, it would ensure all required conditions for the human blood (e.g., temperature, exposure, etc.). All recorded data would be filtered based on privacy regulations and the proposed solutions would be integrated. Furthermore, computer vision and machine learning would help to identify safe landing locations, minimizing the risk of hijacking.

## 4. Requirements, Suggestions, and Evaluation

In this section, we present the necessary prerequisites for the successful deployment of the proposed framework, we discuss possible enhancements, and close with future evaluation plans.

The effective use of the provisioned architecture requires the deployment of a Universal UAV CC (UUCC) that would act as a single point of access for identifying mobile flying objects while they are operating in real environments. The existence of a UUCC would be the first step to establish a well-defined global process for registering and identifying all UAV flights. Additionally, the proposed framework provisions interdrone communication, a fact that requires certain mechanisms to control autonomous flights’ interoperability and co-operation. Interoperability mechanisms set the grounds for defining the types of communications between drones (exchange of data, fleet formulation). For example, the exchange of data between two independent UAVs operating at the same time in two neighbor areas would help in maximizing the area covered, while achieving energy conservation.

The proposed framework, presented in Section 3, supports the management and control of UAV and multi-UAV systems, while preserving the security and privacy of the different stakeholders participating in mobile IoT missions without being differentiated across different application domains. Enhancement of the proposed framework would include variations of the suggested security and privacy mechanisms based on the context of the UAV application and the hardware/mechanical characteristics of the participating UAVs. A classification of the UAVs based on their hardware attributes and their application domain would be particularly useful in filtering security and privacy measures, both at device and network level, ensuring the smooth operation of the mission without overloading the UAV device and the aerial network with procedures that, in some cases, could be out of scope.

The suggested framework, along with the corresponding architecture, is formulated but not yet validated. Therefore, the next logical step would be to build a simulation model as proof of concept of the proposed UAV secure-connectivity framework. Metrics related to security, privacy, and connectivity would be employed in order to evaluate framework and communication effectiveness. Based on the findings, the involved security and privacy techniques would be tuned to adjust to the needs of real-world applications.

## 5. Conclusions

In this survey, the applications of UAVs were reviewed presenting IoT sensors that are essential for the related scenarios and use cases. Considering the drones as IoT devices and the support from emerging technologies such as 5G networks, we analyzed the sensor requirements for the corresponding applications and overview solutions for fleet management over aerial networking. The issues related to privacy and security were presented, focusing on users’ and drones’ privacy. Finally, we proposed a framework that supports and enables these technologies on UAVs, providing advanced security and privacy by incorporating novel vision-based solutions for scene analysis. According to the proposed framework, a hybrid centralized–distributed framework controls UAV flights, handling operations like the registration, identification, ranking, and management of moving objects. As future work, we plan to evaluate the proposed framework, both within laboratory settings and in real-world scenarios, in order to adjust it for context-specific application domains.

## Figures and Tables

**Figure 1 sensors-18-04015-f001:**
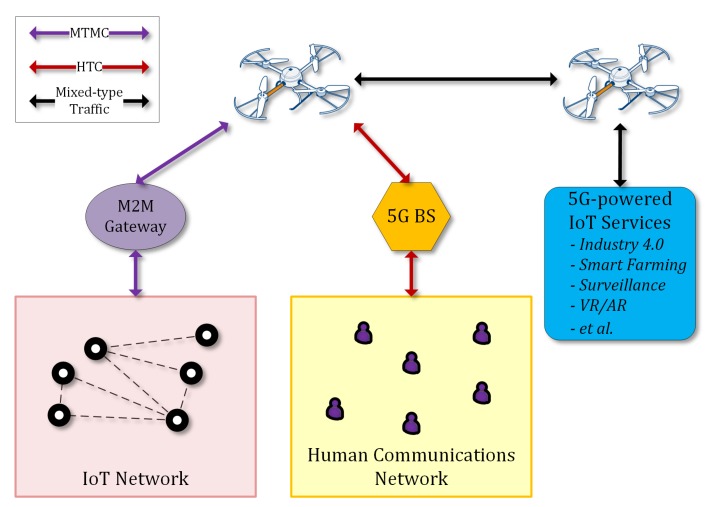
Unmanned aerial vehicle (UAV)-enhanced 5G-enabled Internet of Things (IoT) services.

**Figure 2 sensors-18-04015-f002:**
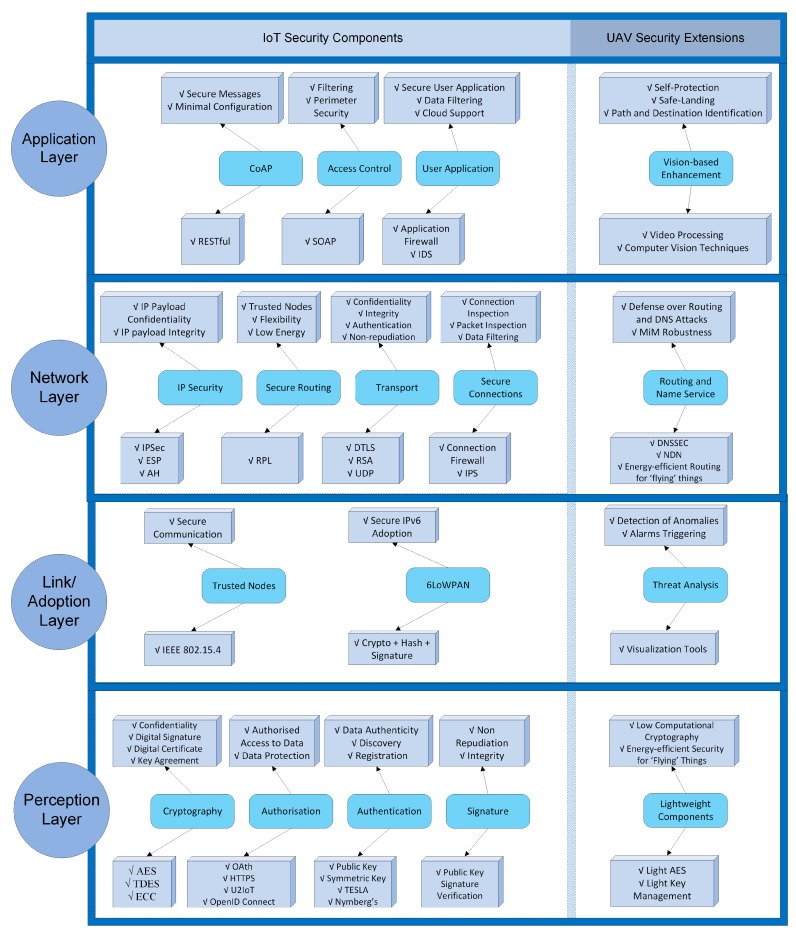
IoT security layers with UAV extensions.

**Figure 3 sensors-18-04015-f003:**
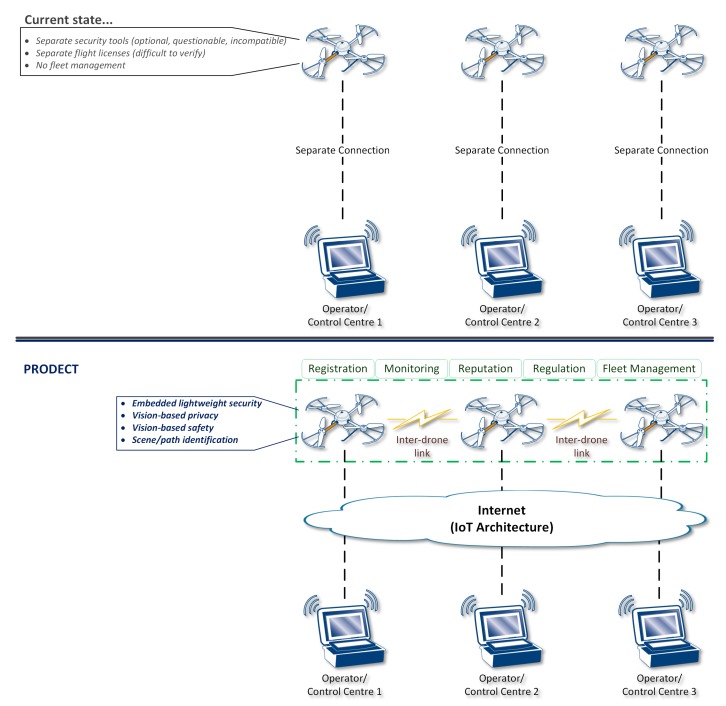
Proposed framework for “flying” things’ secure connectivity in the IoT.

**Figure 4 sensors-18-04015-f004:**
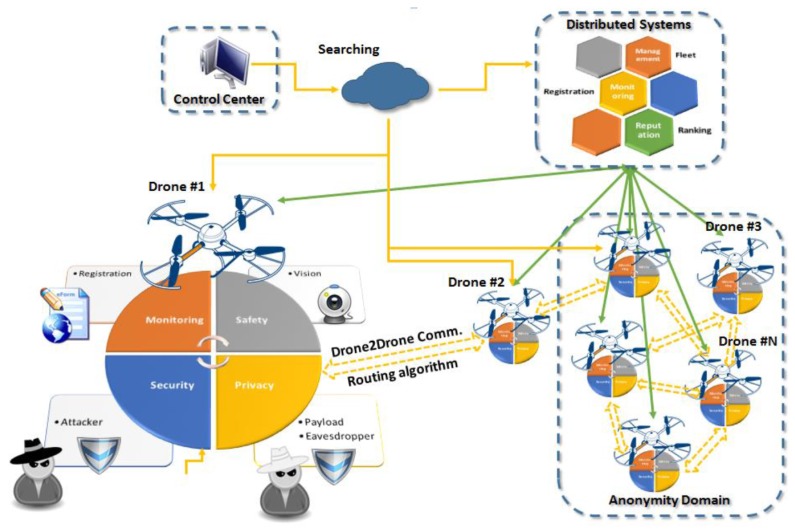
Proposed UAV IoT architecture.

**Figure 5 sensors-18-04015-f005:**
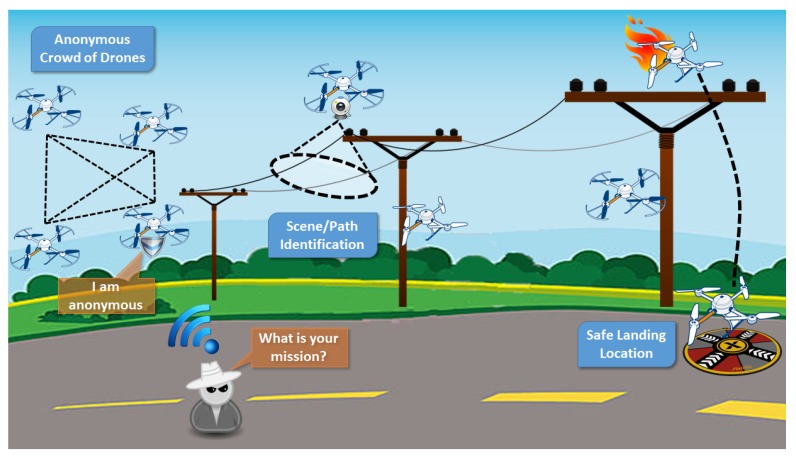
Power-line monitoring application.

**Figure 6 sensors-18-04015-f006:**
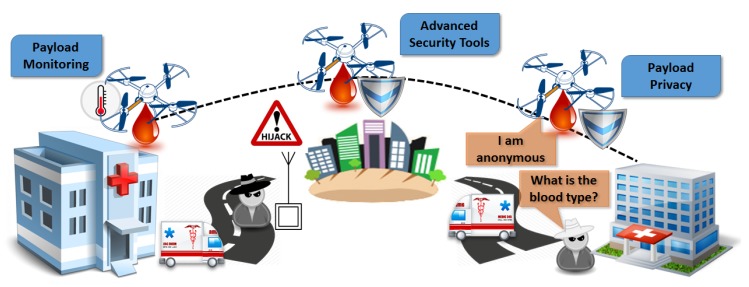
Human blood delivery application.

**Table 1 sensors-18-04015-t001:** Categories of communication technologies available to UAVs.

Category	Technology	Data Rate	Range	Latency
WPAN	Bluetooth 4.0	<1 Mbps	60 m	50
WPAN	Zigbee	<250 kbps	<100 m	50
WLAN	802.11a/b/g/n/ac	<600 Mbps	<250 m	75
WLAN	WAVE 802.11p	<27 Mbps	<1 km	50
LPWA	LoRA	<50 kbps	<15 km	82
LPWA	SigFox	<100 bps	<20 km	82
Cellular	NB-IoT	<250 kbps	World wide	75
Cellular	LTE-M	<1 Mbps	World wide	75
Cellular	LTE Advanced (4G)	<1 Gbps	World wide	50
Cellular	LTE D2D	-	World wide	25
Cellular	5G	<10 Gbps	World wide	3

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
