# Peer review of "UAV IoT Framework Views and Challenges: Towards Protecting Drones as “Things”"

_sensors, 2018, doi:10.3390/s18114015_

Round 1
Reviewer 1 Report
In this paper, the authors study the applications of UAVs as flying things within an IoT architecture sensors that are essential for various scenarios and use cases. While the paper studies a timely UAV-IoT topic, the following comments should be taken into consideration: 1) The main novelty and contributions of the paper need to be clarified. 2) UAVs play key roles in wireless networks. It is important to mention various use case of UAVs, such as flying ad-hoc network, aerial base station, and flying users. 3) Some discussions on the type of UAVs can be added. Also, which types of drones are more appropriate in IoT applications (i.e., flying things). 4) The paper needs to provide some discussion on deployment and path planning of drones in IoT networks. 5) It is important to mention the energy consumption of UAVs and their flight time constraints. In fact, the limited flight time of UAVs is an important issue in UAV systems. 6) Please discuss interference management issue in such UAV-IoT network. Interference is a key challenge due to the massive number of aerial and terrestrial IoT devices. 7) The literature review needs to be significantly improved in this survey. In fact, there are many relevant studies on UAV and IoT communications that need to be mentioned. In particular: [R1] “Unmanned aerial vehicle with underlaid device-to-device communications: Performance and tradeoffs,” IEEE Transactions on Wireless Communications, vol. 15, no. 6, pp. 3949–3963, June 2016. [R2] “Internet of Things for smart cities,” IEEE Internet of Things Journal, vol. 1, no. 1, pp. 22–32, Feb. 2014. [R3] "Efficient Deployment of Multiple Unmanned Aerial Vehicles for Optimal Wireless Coverage", IEEE Communications Letters, vol. 20, no. 8, pp. 1647-1650, Aug. 2016. [R4] “Optimum Placement of UAV as Relays," in IEEE Communications Letters, vol. 22, no. 2, pp. 248-251, Feb. 2018.” [R5]“Mobile Unmanned Aerial Vehicles (UAVs) for Energy-Efficient Internet of Things Communications," IEEE Transactions on Wireless Communications, vol. 16, no. 11, pp. 7574-7589, Nov. 2017. [R6] “A tutorial on UAVs for wireless networks: Applications, challenges, and open problems,” available online: arxiv.org/abs/1803.00680, 2018. [R7] “On the number and 3D placement of drone base stations in wireless cellular networks,” in Proc. of IEEE Vehicular Technology Conference, 2016. [R8] "Resource allocation for machine-to-machine communications with unmanned aerial vehicles." In IEEE Globecom Workshops (GC Wkshps), 2016”.
Author Response
Response to Reviewer 1 Comments
We would like to thank the reviewers for their efforts and their valuable remarks. Accordingly, we have prepared a revised version of the manuscript, which addresses all the comments, resulting in a substantially improved outcome. Below, you may find responses to each one of the comments and explanation of the way they were addressed in the revised version of the paper. All newly added text parts are identified in the revised manuscript in red font colour.
Reviewer 1
In this paper, the authors study the applications of UAVs as flying things within an IoT architecture sensors that are essential for various scenarios and use cases. While the paper studies a timely UAV-IoT topic, the following comments should be taken into consideration:
Comment#1:The main novelty and contributions of the paper need to be clarified.
Response#1: Thank you for the comment. A new paragraph is added in the introduction section summarizing the main contribution and novelty of the paper:
“In summary, the main contribution of this work is on one hand the conduction of a targeted review which focuses on security issues and promising solutions associated with the inclusion of UAVs in the IoT ecosystem, considering the special characteristics of such devices and the related cutting-edge technologies. On the other hand, a new framework that involves UAV-specific security extensions is presented for addressing the identified issues, along with ambitious real-world use cases.”
Comment#2: UAVs play key roles in wireless networks. It is important to mention various use case of UAVs, such as flying ad-hoc network, aerial base station, and flying users.
Response#2: We would like to thank the reviewer for this recommendation. The new subsection 2.1.1 lists use cases for UAVs in wireless networks, based on the recommended reference - A tutorial on UAVs for wireless networks: Applications, challenges, and open problems,” available online: arxiv.org/abs/1803.00680, 2018.
“
- UAV-carried flying base stations that complete heterogeneous 5G systems to enhance the coverage and capacity of existing wireless access technologies.
- UAV-based aerial networks that allow reliable, flexible, and fast wireless connections in public safety scenarios.
- UAVs that support terrestrial networks for disseminating information and enhancing connectivity.
- UAVs as flying antennas that can be deployed on demand to enable mmWave communications, massive MIMO, and 3D network MIMO.
- UAVs that are used to provide energy-efficient and reliable IoT uplink connections.
- UAVs that form the backhaul of terrestrial networks to allow agile, reliable, cost-effective, and high-speed connectivity.
- UAVs able to cache popular content and efficiently serve mobile users by following their mobility patterns.
- UAVs that act as users of the wireless infrastructure for surveillance, remote sensing, virtual reality cases, and package delivery applications.
- UAVs that collect vast amounts of city data and/or enhance its cellular network coverage in a smart city scenario.
”
Comment#3: Some discussions on the type of UAVs can be added. Also, which types of drones are more appropriate in IoT applications (i.e., flying things).
Response#3: Thank you for this suggestion. The new subsection 2.1.2 is added and discusses UAV Types and Classifications and also refer to suitable UAVs for the IoT environment:
“Different types of UAVs with distinctive characteristics, such as supported altitude, speed, and energy autonomy, are suitable for different applications. Generally, UAVs are classified according to their supported altitudes into Low Altitude Platforms (LAP) and High Altitude Platforms [10]. Furthermore, UAVs can be classified into rotary-wing and fixed-wing. The former are appropriate for cases which require UAVs that can remain at steady positions, whereas the latter are suitable for applications that demand UAVs travelling at high speeds and covering large distances [11]. In an IoT environment, due to the limited energy capacity of the participating devices, suitable LAP UAVs of the rotary-wing type can be efficiently and dynamically positioned to allow IoT devices transmit with minimum power [12].”
Comment#4: The paper needs to provide some discussion on deployment and path planning of drones in IoT networks.
Response#4: Thank you for this remark. The new subsection 2.1.3 is added and discusses in two paragraphs deployment and path planning:
“The use of UAVs as flying relays for IoT network has numerous advantages, such as energy conservation and reliability, however, there are also some significant challenges that need to be addressed. Among those challenges, interference management, UAV deployment, and path planning, are considered of major importance. The authors in [14] propose and analyze an efficient deployment scheme for multiple UAVs, using circle parking theory.”
“As far as path planning is concerned, it is directly related with trajectory optimization. In general, finding the optimal flight path for a UAV is considered a challenging goal, since it is affected with multiple factors, such as energy limitations, flight time, and obstacle avoidance. Hence, as explained in [9], path planning is usually approached as an optimization problem with various objectives, depending on the criterion of interest.”
Comment#5: It is important to mention the energy consumption of UAVs and their flight time constraints. In fact, the limited flight time of UAVs is an important issue in UAV systems.
Response#5: We would like to thank the reviewer for this recommendation. The new subsection 2.1.3 is added and discusses in a paragraph energy consumption issues:
“Energy consumption, in particular, constitutes a critical issue for the deployment and mobility of UAVs. Because of their limited battery capacity, UAVs are not typically able of providing for long continuous wireless coverage, in scenarios such as IoT networking. Their energy autonomy is highly affected by the UAV role, the flight path, the weather conditions et al. and actually constitutes the main constraint for UAV adoption in many cases. There are several recent research endeavours towards improving UAV energy efficiency, focusing on various aspects, such as trajectory optimization [17], cooperative communications [18], energy harvesting [19], resource allocation [20], et al.”
Comment#6: Please discuss interference management issue in such UAV-IoT network. Interference is a key challenge due to the massive number of aerial and terrestrial IoT devices.
Response#6: Thank you for the comment. The new subsection 2.1.3 is added and discusses in a paragraph interference management issues:
“Regarding interference management, the findings revealed that UAVs' altitude needs to be adjusted according to the coverage requirements and the beamwidth of their directional antennas. A related work presented in [15] concluded about the optimum placement of UAVs as relay nodes that the decode-and-forward approach outperforms the amplify-and-forward one. A new heuristic algorithm for 3D UAV deployment was introduced in [16], which minimizes the number of required UAVs to keep a specific level of service quality. To mitigate interference, the authors suggest lowering the altitude, but there is an obvious tradeoff between this and coverage.”
Comment#7: The literature review needs to be significantly improved in this survey. In fact, there are many relevant studies on UAV and IoT communications that need to be mentioned. In particular:
[R1] “Unmanned aerial vehicle with underlaid device-to-device communications: Performance and tradeoffs,” IEEE Transactions on Wireless Communications, vol. 15, no. 6, pp. 3949–3963, June 2016.
[R2] “Internet of Things for smart cities,” IEEE Internet of Things Journal, vol. 1, no. 1, pp. 22–32, Feb. 2014.
[R3] "Efficient Deployment of Multiple Unmanned Aerial Vehicles for Optimal Wireless Coverage", IEEE Communications Letters, vol. 20, no. 8, pp. 1647-1650, Aug. 2016.
[R4] “Optimum Placement of UAV as Relays," in IEEE Communications Letters, vol. 22, no. 2, pp. 248-251, Feb. 2018.”
[R5]“Mobile Unmanned Aerial Vehicles (UAVs) for Energy-Efficient Internet of Things Communications," IEEE Transactions on Wireless Communications, vol. 16, no. 11, pp. 7574-7589, Nov. 2017.
[R6] “A tutorial on UAVs for wireless networks: Applications, challenges, and open problems,” available online: arxiv.org/abs/1803.00680, 2018.
[R7] “On the number and 3D placement of drone base stations in wireless cellular networks,” in Proc. of IEEE Vehicular Technology Conference, 2016.
[R8] "Resource allocation for machine-to-machine communications with unmanned aerial vehicles." In IEEE Globecom Workshops (GC Wkshps), 2016”.
Response#7: Thank you for the suggested references. All eight of them (and more) are now added in the revised version to enhance the literature review.

Reviewer 2 Report
This manuscript proposes a UAV-based IoT framework with two specific examples of the power line monitoring and human blood delivery, by investigating the recent researches on sensor technology, security, and protection. The type of this manuscript is a review paper; however, it is mainly focused on the proposed UAV-based IoT framework rather than the introduction of the prior studies, as shown in Sections 3 and 4. The reviewer recommends the authors to revise the manuscript with the following comments.
Comments are as follows:
1. In Section 2, the overview of UAVs for the IoT framework is organized as four sub-sections, including “2.1 IoT sensors for UAVs over 5G networks”, “2.2 Sensor technologies for UAVs”, “2.3 Security for UAVs over IoT”, and “2.4 Protection for UAVs”.
a. Before starting Sub-section 2.1, a brief introduction and schematic demonstrations are needed for better understanding.
b. The contents in Sub-sections 2.1 and 2.2 are seen to be similar, because they are talking about the sensor technologies for UAVs.
c. In Sections 2.3 and 2.4, the security and protection for UAVs are discussed. Are the mentioned references in these sections related to UAVs? In other words, are these approaches being adopted in practical applications with UAVs?
2. The proposed UAV-based IoT framework and corresponding discussions take up too much of this manuscript (see Sections 3 and 4). The reviewer recommends the authors to provide more prior studies related to the practical application of UAVs with IoT frameworks.
Author Response
Response to Reviewer 2 Comments
We would like to thank the reviewers for their efforts and their valuable remarks. Accordingly, we have prepared a revised version of the manuscript, which addresses all the comments, resulting in a substantially improved outcome. Below, you may find responses to each one of the comments and explanation of the way they were addressed in the revised version of the paper. All newly added text parts are identified in the revised manuscript in red font colour.
Reviewer 2
This manuscript proposes a UAV-based IoT framework with two specific examples of the power line monitoring and human blood delivery, by investigating the recent researches on sensor technology, security, and protection. The type of this manuscript is a review paper; however, it is mainly focused on the proposed UAV-based IoT framework rather than the introduction of the prior studies, as shown in Sections 3 and 4. The reviewer recommends the authors to revise the manuscript with the following comments.
Comments are as follows:
Comment#1: In Section 2, the overview of UAVs for the IoT framework is organized as four sub-sections, including “2.1 IoT sensors for UAVs over 5G networks”, “2.2 Sensor technologies for UAVs”, “2.3 Security for UAVs over IoT”, and “2.4 Protection for UAVs”.
Comment#1a: Before starting Sub-section 2.1, a brief introduction and schematic demonstrations are needed for better understanding.
Response#1a: Thank you for this comment. A new subsection 2.1 is now added before the old subsection 2.1 (now 2.3), which explains the role of UAVs in wireless networking. Moreover, a new Figure 1 is added to illustrate the use of UAVs for enhancing 5G-enabled IoT services (integrating MTMC and HTC traffic).
Comment#1b: The contents in Sub-sections 2.1 and 2.2 are seen to be similar, because they are talking about the sensor technologies for UAVs.
Response#1b: Thank you for this remark. These sub-sections are now merged in 2.3 under the heading “5G and IoT Sensor Technologies for UAVs”
Comment#1c: In Sections 2.3 and 2.4, the security and protection for UAVs are discussed. Are the mentioned references in these sections related to UAVs? In other words, are these approaches being adopted in practical applications with UAVs?
Response#1c: Thank you for this comment. The reference used in those two sections are related to IoT-oriented techniques and/or to UAV issues. A key objective of this work is to combine and extend related technologies in order to secure UAVs as part of the IoT environment. This is now clarified in the beginning of section 2.4:
“Security provision in UAVs as part of the IoT environment is a complex task that requires the efficient integration of various techniques, which are associated with different aspects of IoT networking and UAV operation. In the following two subsections, the security and privacy components for such an endeavour are discussed in detail. The main concept of the followed approach is the application of UAV-specific security extensions to the various IoT technologies and security techniques, which can enable the required integration.”
Comment#2: The proposed UAV-based IoT framework and corresponding discussions take up too much of this manuscript (see Sections 3 and 4). The reviewer recommends the authors to provide more prior studies related to the practical application of UAVs with IoT frameworks.
Response#2: Thank you for this recommendation. A new section “2.2 UAV-IoT Frameworks” is now added which presents and discusses prior studies related to the practical application of UAVs with IoT frameworks. The corresponding text follows:
“Due to UAVs' high agility, they are now widely accepted as promising members of the IoT vision or even enablers of such a vision. They are capable of offering new value-added IoT services, while they can carry a variety of MTMC devices [21]. In more detail, according to the definition of IoT, "things" are expected to be able to be connected anywhere at anytime providing any service. UAVs can fulfil this requirement, thanks to their autonomy, flexibility, and programmability. In this context, a number of UAV-enabled IoT frameworks supporting a variety of practical use cases have been proposed.
Authors in [1] introduced and demonstrated a UAV-based IoT platform for crowd surveillance. The respective platform adopts and applies face recognition techniques and performs efficient offload of video processing to a Mobile Edge Computing (MEC) node, considering the limited processing power and energy capacity of a UAV. The developed testbed, collects video surveillance data and performs face recognition to identify suspicious individuals utilizing the Local Binary Pattern Histogram (LBPH) algorithm of the Open Source Computer Vision (OpenCV) library. The proposed platform considers central management of a fleet of UAVs through a system orchestrator.
A communication framework for UAVs in urban IoT environments was proposed and evaluated in [22]. It forms a multi-path multi-hop infrastructure which is used to connect the UAVs to the Ground Control Station. The conducted real-world experiments have shown that the introduced framework significantly enhances the control effectiveness and reliability against local congestion. It is noted that the specific work was inspired by the DARPA Hackfest on Software Defined Radios.
In [23], a game-theory based framework was introduced for allocating resources to UAVs, which enter the IoT ecosystems as platforms that assist terrestrial base stations. The access competition among the UAVs for bandwidth is modelled as a non-cooperative evolutionary game. The evaluation of the two designed algorithms showed that Nash equilibrium can be reached fast.
An optimization framework for aerial sensing in the context of an IoT infrastructure was designed and presented in [24]. The goal is to allow remote users navigate in specific scenes of interest by using AR/VR devices over the captured data. The corresponding scenario is likened to virtual human teleportation. The conducted experiments effectively demonstrated the advantages of the proposed methods on visual sensing.
Authors in [25] conceived and presented a new MEC framework for IoT through an air-ground integration approach. Four use cases are presented to show how the proposed air-ground integrated MEC framework support high mobility, low latency, and high throughput for 5G applications. Through simulation-based and case=based evaluation, it was shown that the respective framework can support multiple IoT scenarios.
A novel framework for deploying and efficiently moving UAVs to gather information from ground IoT devices is proposed in [26]. This work focuses on optimal deployment and mobility of UAVs, as well as optimal clustering of IoT devices, towards minimizing transmission power, while retaining reliability. In this manner, it was shown that IoT devices energy consumption can be significantly reduced, whereas UAVs can serve ground devices for longer time.”

Round 2
Reviewer 1 Report
I would like to thank the authors for addressing most of my previous comments. While the quality of the paper has been substantially improved, the following minor comments still need to be taken into account:
1) The following relevant works have not been mentioned in the literature review:
[R1] “Unmanned aerial vehicle with underlaid device-to-device communications: Performance and tradeoffs,” IEEE Transactions on Wireless Communications, vol. 15, no. 6, pp. 3949–3963, June 2016.
[R2] “Internet of Things for smart cities,” IEEE Internet of Things Journal, vol. 1, no. 1, pp. 22–32, Feb. 2014.
2) Please increase the font-size of small texts in Figure 3.
Author Response
Response to Reviewer 1 Comments – Round2
I would like to thank the authors for addressing most of my previous comments. While the quality of the paper has been substantially improved, the following minor comments still need to be taken into account:
Comment#1: The following relevant works have not been mentioned in the literature review:
[R1] “Unmanned aerial vehicle with underlaid device-to-device communications: Performance and tradeoffs,” IEEE Transactions on Wireless Communications, vol. 15, no. 6, pp. 3949–3963, June 2016.
[R2] “Internet of Things for smart cities,” IEEE Internet of Things Journal, vol. 1, no. 1, pp. 22–32, Feb. 2014.
Response#1: Thank you for pointing this out. In the second revision, we have included these two references.
Comment#2: Please increase the font-size of small texts in Figure 3.
Response#2: Thank you for the recommendation. We have now increased the overall size of the specific figure and the small fonts.
